# Bacterial Colonization of Irrigation Fluid during Aseptic Revision Knee Arthroplasty

**DOI:** 10.3390/jcm9092746

**Published:** 2020-08-25

**Authors:** Michael Fuchs, Matthias Pumberger, Hagen Hommel, Carsten Perka, Philipp von Roth, Kathi Thiele

**Affiliations:** 1RKU University Department of Orthopaedics, University of Ulm, 89081 Ulm, Germany; 2Center for Musculoskeletal Surgery, University Department of Orthopaedics, Charité–University Medicine, 10117 Berlin, Germany; matthias.pumberger@charite.de (M.P.); carsten.perka@charite.de (C.P.); kathi.thiele@charite.de (K.T.); 3Department of Orthopaedics, Märkisch-Oderland Hospital, 16269 Wriezen, Germany; H.Hommel@khmol.de; 4Sporthopaedicum Regensburg and Straubing, 93053 Regensburg, Germany; roth@sporthopaedicum.de

**Keywords:** surgical contamination, irrigation liquid, fluid reservoirs, total knee replacement

## Abstract

Surgical risk factors for periprosthetic joint infection (PJI) after total knee arthroplasty (TKA) are the subjects of ongoing research. It is unclear if there are specific locations of the surgical area that might act as a pathogen source. Due to the fact that bacterial replication occurs preferably under humid conditions, it was our aim to investigate if irrigation fluid reservoirs on the surgical covers are subject to bacterial colonization. We prospectively observed 40 patients with scheduled aseptic 1-stage TKA revision. At time intervals of 30 min, irrigation samples were tested for microbiological colonization. Additionally, the suction tip was investigated at the end of surgery. Overall, a bacterial detection rate of 25% was found (57/232 samples). Analysis for any positive microbial detection revealed pathogen findings of irrigation fluid in 41.7% of samples after 30 min with a constant increase up to 77.8% after 90 min. Twenty-three percent of suction tips showed bacterial colonization. Coagulase-negative staphylococci, accounting for the majority of PJI, were the predominant pathogens. After an average follow-up of 17 months, no PJI was confirmed. Despite the substantial bacterial load of irrigation fluid, PJI rates were not elevated. Nevertheless, we recommend that irrigation fluid reservoirs should be prevented and not withdrawn by suction.

## 1. Introduction

Periprosthetic joint infection (PJI) is one of the most devastating complications after total knee arthroplasty (TKA). Despite improved surgical techniques, guideline-based antibiotic prophylaxis and individual patient conditioning, the infection rate in revision TKA constantly remains between 3–38% [1,2,3,4,5,6]. With regard to its etiology, patient and surgical factors must be differentiated. So far, many studies focused on patient specific parameters leading to a higher susceptibility of PJI.

Patient factors that are associated with deep surgical site infection include a body mass index (BMI) of ≥35, diabetes mellitus, an American Society of Anesthesiologists score of ≥3, and a diagnosis of posttraumatic arthritis [7]. With regard to surgical risk factors, an extensive surgical site exposure as well as the length of the operation are related to increased infection rates following TKA [7]. With reference to surgical instrument contamination, different studies showed a distinct bacterial charge of suction tips, surgical gloves, and fluid collection bags [8,9,10]. Givissis et al. reported a bacterial colonization rate of 54% for suction tips in the course of elective orthopedic and trauma procedures in 50 patients [9]. In 2005, another study investigated glove perforation and contamination in 627 pairs of outer surgical gloves in primary total hip arthroplasty (THA). The authors concluded that bacterial contamination takes place in up to 14% and gave advise that glove changing at regular intervals is effective in decreasing the overall pathogen load [8]. For fluid collection bags used in primary THA, a bacterial colonization of the forming reservoirs of almost 10% has been described [10]. Interestingly, PJI rates were not elevated over all in the observed patient cohorts. Thus, there is still a lack of evidence with regard to surgical parameters and their impact on implant colonization.

One key procedure to avoid intraoperative bacterial colonization is constant irrigation of the surgical site. As shown in previously published animal studies, irrigation of bacterial contaminated wounds substantially reduced the local microbial load [11]. Although, by redundant irrigation, a certain degree of pathogen dilution is undertaken, it results in fluid reservoirs on the sterile surgical drapes. The latter might not only consist of sterile infusion solution, blood, and debris but also could represent a potential niche for bacteria, as those pathogens preferably replicate under humid and warm conditions. With regard to the complex and long-lasting procedure of TKA revision surgery, in which various surgical instruments are used and redundant irrigation of the surgical field is undertaken, this issue may be of particular importance.

Against this background, a question arises: Do fluid reservoirs and suction tips represent a potential source of infection and thus pose a threat for PJI development? To clarify this, the aim of this study was to evaluate if irrigation fluid reservoirs and disposable suction tips are subject to bacterial colonization within the time course of aseptic revision knee arthroplasty.

## 2. Material and Methods

### 2.1. Study Design and Patients

This prospective study investigated the microbiological colonization of irrigation fluid and the suction tip in revision TKA procedures. Data were collected from April 2017 to January 2018. The study protocol was approved by the local ethics committee (registration number: S 20(a)/2015). Forty consecutive patients who underwent aseptic revision TKA were included after prospective identification and fulfillment of strict inclusion and exclusion criteria. Aseptic TKA failure leading to revision surgery was diagnosed by systemic blood sampling, joint aspiration, and clinical presentation without redness or swelling of the affected limb. The inclusion criterion was a minimum age of 18 years, scheduled aseptic 1-stage revision surgery of either the femoral, tibial, or both implant components, negative microbiological testing during revision surgery, and written informed consent to participate in the study. The exclusion criterion was revision surgery due to PJI, according to the European Bone and Joint Infection Society (EBJIS) criteria [12,13]. According to these definitions, PJI was diagnosed when at least one of the following criteria were present: (1) macroscopic purulence around the prosthesis; (2) presence of a sinus tract; (3) an increased leucocyte cell count in synovial fluid (>2000 leucocytes/µL) or a differential count of >70% granulocytes; (4) relevant microbiological growth in synovial fluid, prosthetic tissue, or sonication culture of the retrieved components; (5) positive histopathology, defined as ≥23 granulocytes per ten high-power fields, corresponding to a type II or III classification of the periprosthetic membrane according to Morawitz and Krenn [14]. Low-virulent microorganisms such as *coagulase-negative staphylococci* or *Propionibacterium* species were considered relevant if the same organism was isolated in at least two samples or in one sample if at least one other criteria for the diagnosis of PJI was present.

### 2.2. Revision Surgeries

The indications for TKA revision surgery were joint instability or aseptic implant loosening. All patients received standard radiological examinations before revision surgery (standing long leg anterior–posterior (AP) x-ray as well as AP and lateral x-ray knee joint projection). In the case of an unclear radiological examination, computed tomography (CT) imaging was performed to confirm implant loosening. Joint instability was assessed via AP and sagittal stress radiographs. Three experienced, high-volume surgeons at the senior level performed all revision procedures. All patients received preoperative skin decolonization following a standard protocol. Therefore, patients had to use a decolonization kit with octenidin prepared hygienic disposable washcloths as well as an octenidin nasal ointment (Octenisan, Schülke & Mayr, Norderstedt, Germany). Patients were asked to wash their whole body on the preoperative day with the disposable washcloths. On the day of surgery, only the surgical site was decolonized. Nasal ointment was used from the preoperative day until the third postoperative day, twice daily. The surgeons wore sterile, disposable operating room clothing. Standard preoperative skin scrubbing of the patient’s lower extremity was performed using an antiseptic solution (Softasept N; B. Braun, Melsungen, Germany). For skin disinfection of the surgical site, the skin was washed four times with an antiseptic solution (Braunol; B. Braun, Melsungen, Germany) for at least 10 min. Patients received single-shot perioperative antibiotic prophylaxis (cefazoline 2 g intravenously 30 min before skin incision). In the event of cephalosporin intolerance, 1 g of vancomycin was administered intravenously 2 h before skin incision. For procedures with a surgical time beyond 2 h, a second single-shot antibiotic prophylaxis was administered after 120 min. The median parapatellar approach was used as the standard surgical approach in all cases. There was a laminar airflow and outwardly directed excess pressure during the entire procedure. All patients underwent cemented revision TKA. In all patients, intravenous tranexamic acid (TXA) administration (1 g) as well as intraarticular application of TXA 2 g after closure of the joint capsule were performed to reduce postoperative bleeding complications. No postoperative antibiotics were given in any of the included cases.

### 2.3. Study Parameters and Sample Collection

During surgery, irrigation fluid was collected and tested for bacterial contamination after defined time intervals of 30 min (Figure 1). A crystalloid sodium chloride solution (NaCl 0.9%; B. Braun, Melsungen, Germany) was used as the irrigation solution with a 50 mL syringe and a fluid irrigation device (Pulsavac, Zimmer, Warsaw, IN, USA). At the beginning of surgery, a control sample was taken from the irrigation container to exclude initial contamination (negative control). Sterile draping was performed following a routine protocol, with a two-layer sterile covering towel. Before skin incision, 20 mL of irrigation fluid was poured onto the surgical covers. Of these, 3 to 5 mL were immediately collected from the surgical drapes with a sterile 10 mL syringe for microbiologic evaluation. Every 30 min and at the end of surgery, 3 to 5 mL of irrigation fluid from the surgical drapes were investigated (Table 1). All samples were injected into blood culture bottles (Bactec Peds- Plus; Becton Dickinson, Franklin Lakes, NJ, USA) and incubated for 14 days. In addition, at the end of surgery, the disposable suction tip (Dahlhausen Medical equipment; Cologne, Germany) was retained for subsequent microbiologic testing: The distal 3 cm of the tip were cut and immediately incubated in a 50 mL Falcon Tube (Fisher Scientific, Schwerte, Germany) with 32 mL thioglycolate broth. After concentration for 48 h, 10 µL were extracted and plated on agar plates under aerobic and anaerobic conditions. After surgery, all samples were sent to a microbiologic institute where they were incubated for 14 days. In case of pathogen findings, bacterial species were identified and resistograms were established. For reasons of clarity, bacterial analysis was illustrated as follows and according to previously published studies: *Coagulase negative staphylococci* were differentiated in *S. lugdunensis*, which can be seen as a high virulent pathogen and *other coagulase negative staphylococci* (*CoNS*) with a low virulent potential. Furthermore, *micrococci*, *other gram-positive bacteria*, *gram-negative bacteria*, and *anaerobes* were distinguished [15,16,17].

### 2.4. Follow-Up

After discharge from our clinic, follow-up examinations were scheduled on a routine basis after 3 and 12 months for clinical and radiological examination. After 12 months, patients were seen annually. In the case of painful TKA, PJI was excluded by systemic blood sampling, joint aspiration, and radiological examination. In patients without complaints, PJI was excluded by systemic blood sampling, clinical examination, and radiographic imaging.

### 2.5. Statistical Analysis

The Mann-Whitney Test was used to compare unpaired variables between groups. Results were illustrated as mean and standard deviation (SD) or as number and percentage. The prediction analysis is specified with a 95% prediction interval. Statistical analysis was performed using GraphPad Prism Version 8.0 (GraphPad Software, San Diego, CA, USA) and SPSS (IBM SPASS Statistics 25, Armonk, NY, USA). A *p*-value < 0.05 was considered significant.

## 3. Results

### 3.1. Demographic Data

According to confirmed PJI in sonication-based and microbiological analyses of the removed implants during revision TKA, three patients had to be excluded from further evaluation. Additionally, one patient had to be excluded due to a lost irrigation fluid sample. Thus, a total of 36 patients, including 23 women and 13 men, were evaluated. The average age of patients at the time of revision surgery was 66.5 years (range, 50–81 years, SD 9 years), BMI measurements revealed an average index of 31.6 (range, 21.6–45.9, SD 6.6). Mean surgical time was 131 min (range, 79–250 min, SD 35.7 min).

### 3.2. Microbiological Analysis

In total, 232 samples (197 irrigation fluid samples and 35 suction tip samples) were collected at the respective time points, which consisted of the following: t-0, negative control at the beginning of surgery (*n* = 36); t-0, irrigation fluid sample at the beginning of surgery (*n* = 36); t-30, irrigation fluid sample after 30 min (*n* = 36); t-60, irrigation fluid sample after 60 min (*n* = 36); t-90, irrigation fluid sample after 90 min (*n* = 32); t-120, irrigation fluid sample after 120 min (*n* = 21); 35 suction tip samples at the end of surgery. One suction tip sample was lost during transportation. In total 25% (*n* = 57) of all samples were positive for microbiological colonization.

One (3%) of the negative control samples taken from the irrigation container at the beginning of surgery turned out to be positive. In two patient samples (6%) taken before incision, bacterial colonization was detected. After 30 min, 13 of 36 patients´ samples were colonized (36%). At 60 min, 16 of 36 patients’ samples tested positive for microbial pathogens (44% of samples). By min 90, bacterial findings were observed in 13 of 32 patients’ samples (41%). After 120 min, 4 out of 24 samples (19%) showed microbiological colonization. The suction tip investigation at the end of the surgery showed bacterial colonization in 8 patients (23%, Figure 1). Mean surgical times did not differ between patients with positive culture results (132 min, SD 39 min) and those with sterile irrigation fluid reservoirs (129 min, SD 17 min, *p* = 0.98). For all detected pathogens, bacterial resistograms revealed sufficient susceptibility to the administered antibiotic prophylaxis. More specifically, no multi-resistant bacterial strains were observed.

With regard to the cumulative colonization rate of any positive microbial finding during surgery, there was a steady increase from 41.7% after 30 min to 77.8% at the end of surgery (Figure 2). The detected pathogens were differentiated into 6 groups: *other coagulase negative staphylococci* (*CoNS*, *n* = 53), *micrococci* (*n* = 8), *gram-positive rods* (*n* = 6), *staphylococcus lugdunensis* (*n* = 2), *anaerobes* (*n* = 1), and *gram-negative rods* (*n* = 1) (Figure 3). Detailed analysis of *CoNS*, which were the predominant bacterial colonization pathogens of the irrigation fluid, showed an appearance of *S. epidermidis* in 44% of samples and were the most common type of this bacterial strain. Suction tips were also most often colonized by *CoNS* (78%), followed by *anaerobes* (22%).

### 3.3. Follow-Up

Of the 36 included individuals, no patient was lost to follow-up. After an average follow-up of 17 months (range, 12–24 months, SD 5 months), no clinical or laboratory signs of PJI were observed at the scheduled examinations at our outpatient department. Two patients with painful TKA underwent joint aspiration without bacterial detection or elevated leucocyte cell counts suspicious for infection according to the EBJIS criteria. Thus, no PJI was confirmed within the follow-up period. With regard to radiological evaluations, no implant loosening was evident at the respective examinations after 3, 12, and 24 months. No revision surgeries were performed within the follow-up period.

## 4. Discussion

Despite the profound knowledge on patient-related risk factors contributing to PJI, there is a lack of evidence highlighting surgical parameters and their impact on implant colonization. Especially in the course of TKA revision surgery, in which various surgical instruments are used and redundant irrigation of the surgical field is undertaken, the forming fluid reservoirs might pose a threat for bacterial charge. Against this background, the latter were the subject to the present study, in which an overall bacterial detection rate of 25% (57/232 samples) was found in 197 irrigation and 35 suction tip samples. Analysis for any positive pathogen detection revealed a microbial finding of irrigation fluid in 41.7% of samples after 30 min with a constant increase up to 77.8% after 90 min. *Coagulase-negative staphylococci*, counting for the majority of PJI, were the predominant pathogens [18,19]. Although a high pathogen load of irrigation fluid reservoirs was observed, these findings were not associated with an elevated risk of PJI. To our best knowledge, there is no comparable study investigating irrigation fluid contamination and PJI after aseptic revision TKA in the literature.

Previous research focused on the colonization of supposedly sterile surgical equipment such as gloves, splash basins, irrigation fluid, incise draping devices, or suction tips during surgery [8,9,16,20,21,22]. Despite substantial bacterial contamination of those specimens, no elevated risk of periprosthetic infections could be determined overall. In 2018, it was shown that irrigation fluid reservoirs on sterile surgical drapes bear a high risk of bacterial colonization in primary TKA procedures [16]. While there were almost no bacterial findings within 30 min, microbiological analysis revealed positive findings in 22% of samples after 60 min of surgical time. Compared to our previous work, we obtained a slightly higher bacterial detection before skin incision of 6% (t0). After 30 min, there was a strong increase of microbiological findings (36%), indicating an earlier and more excessive rise than described before (22% at 60 min) [16]. Theoretically, the mechanical work associated with the explantation of TKA devices goes along with a more extensive stress of the surrounding soft tissues. Given this thought, a potential bacterial release of skin appendages might be conceivable. However, we again were able to show a constant increase of positive pathogen findings over time. With regard to the analyzed suction tips, a very similar colonization rate of 23% compared to our previous work (22%) was observed. Interestingly, a substantial percentage decrease of positive results between 90 min (41%) and the end of surgery (19%) was evident. An explanation for this might be an excessive pulsatile jet lavage and irrigation of the bone surfaces before implant cementation. The latter is performed immediately before joint closure and requires a clean bone bed achieved by meticulous irrigation. In this context, we speculate that excessive jet lavage preparation leads to a substantial dilution of the irrigation fluid reservoirs and therefore to a reduced pathogen load. Furthermore, there was a second antibiotic administration after 120 min in patients with a surgical time beyond 2 h. Though this remains highly speculative, the bactericidal effects of the second antibiotic treatment may have contributed to the observed decrease of bacterial colonization.

With regard to pathogen-specific evaluation, a huge variety of *coagulase negative staphylococci* were detected. Among these, *S. epidermidis*, which is one of the most frequent pathogens leading to PJI [1], was determined in 44% of samples. Our findings suggest that the vast majority of bacterial findings can be traced back to the patient’s skin flora [8,20,22]. Though there is no evidence that antiseptic incision skin draping has the potential to reduce septic complications, further studies could focus on a potential reduction of bacterial fluid reservoirs with these devices [23]. In this context, the question arises if incision drapes could prevent irrigation fluid from bacterial colonization, as they may act as an artificial skin border, protecting the fluid medium from potential skin pathogens. Beldame et al. analyzed the perforation and pathogen colonization of surgical gloves in 28 patients after primary THA. Although the authors found a comparable high pathogen detection rate of 54%, again, no infection-related complications could be identified after a 12-month follow-up period [20]. Anto et al. evaluated the contamination of splash basin liquid in primary TKA and total hip arthroplasty (THA) procedures. *Pseudomonas* and *CoNS* were detected in 24% of the surgeries with an average operating time of 96 min. As a result of these findings, the authors recommended to avoid surgical instrument storage in splash basins [24]. Potential colonization of suction tips was previously investigated by several authors, whereby the extent of colonization seems to be negatively influenced by the type (elective versus trauma surgery) and duration of surgery [9,16,21]. This implies that although there is no sterile surgical environment within the course of an operation, an intra-articular transmission leading to PJI cannot be automatically concluded. The following thoughts are linked to this: On the one hand, perioperative administered antibiotics seem to bear the potential to eliminate pathogens sufficiently. On the other hand, the question arises if there is a certain kind of patient predisposition, in terms of immunological competence, leading to endogenous pathogen resilience.

Concerning the limitations of our study design, it has to be noted that the 36 interventions were associated with different operative times, which in turn correlate with the colonization rate. In addition, the evaluated follow-up may not be sufficient to record all potential septic complications. This aspect is particularly important for low-grade infections, as they are usually diagnosed after a longer follow-up period and thus the final rate of related PJI cannot yet be reliably determined.

## 5. Conclusions

With the results of this study, we confirm the high bacterial colonization rate of irrigation fluid reservoirs in aseptic revision TKA. We furthermore state that the vast majority of pathogens are dissolved from the patient’s skin and, thus, redundant skin contact should be reduced to a minimum. According to our results, irrigation fluid reservoirs should not be withdrawn by suction in order to minimize a potential bacterial source of infection. In addition, our study underlines the importance of repeated irrigation in order to achieve a high degree of pathogen dilution. The main emphasis with regard to pathogen reduction focuses on the prevention of irrigation fluid reservoirs. Therefore, devices such as arthroscopic suction bags may reduce surgical-site bacterial load.

## Figures and Tables

**Figure 1 jcm-09-02746-f001:**
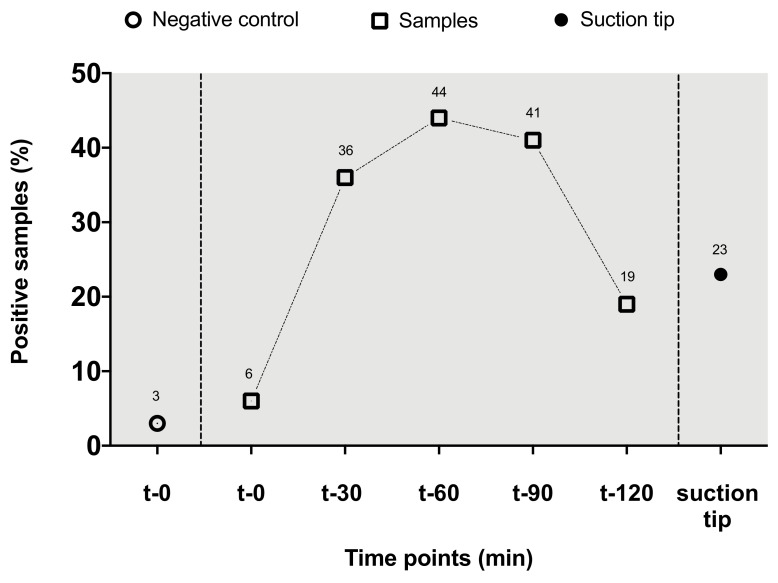
Percentage of positive samples per time point (depicted above representative symbols). Negative control, sodium chloride bag.

**Figure 2 jcm-09-02746-f002:**
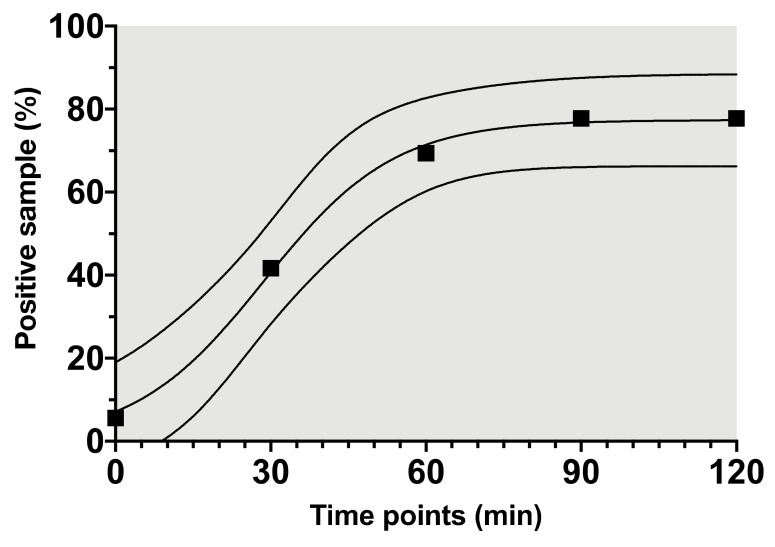
Individual cumulative risk of a positive irrigation fluid pathogen finding over time. 95% prediction interval presented as dashed lines; R square 0.99.

**Figure 3 jcm-09-02746-f003:**
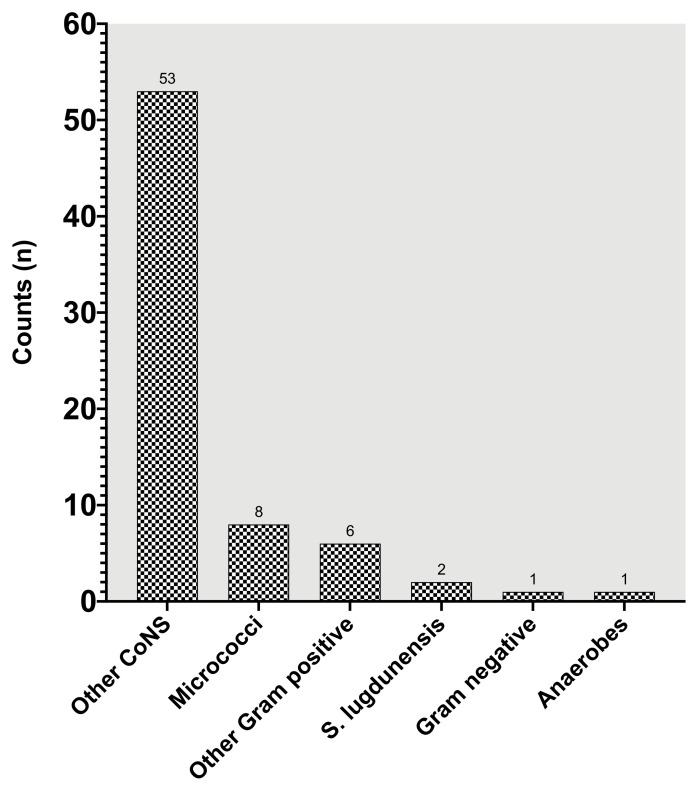
Differentiation of bacterial strains. Data are presented as quantitative bacterial counts.

**Table 1 jcm-09-02746-t001:** Documentation of the sampling schedule according to sampling site and time point

Sampling Site	Samples Taken					
	t 0 min	t 30 min	t 60 min	t 90 min	t 120 min	Endpoint
Sodium-chloride solution bag	x					
Surgical covers	x	x	x	x	x	x
Suction tip						x

Counting (0 min) starts with skin incision; t, time point; time interval = 30 min; endpoint = time of skin closure.

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
