# Peer review of "Bacterial Colonization of Irrigation Fluid during Aseptic Revision Knee Arthroplasty"

_jcm, 2020, doi:10.3390/jcm9092746_

Round 1
Reviewer 1 Report
Excellent manuscript with the infection risk being a critical aspect of a TKA revision setting. Based on the risk factors, the revision setting, its important to know the risks associated with a infection profile. having the bacterial location is critical to understand the nature of the infection profile and location to avoid and or minimize the risks. Well written manuscript
Reviewer 2 Report
Thank you very much for allowing me to read your work.
Overall I believe that this is an interesting study although the implications for everyday Patient care are somewhat unclear as there were no PJIs in your cohort.
Furthermore, I would require a Minimum follow-up of 24 months which is usually the Standard for reconstructive procedures.
I would like to see you comment on the use of antiseptic irrigations intraoperatively, the use of an antiseptic incisional draping or the use of preroperative Skin decolonisation.
Were there any Revision surgeries?
Furthermore, to me it would be interesting to see the resistance testing of the Bacteria identified and whether they were susceptible to the perioperative antibiotics. Otherwise your conclusions cannot be hold up.
Reviewer 3 Report
My overall impression is that the paper is well written, focusing on an interesting topic, but it requires higher evidence and stronger conclusions, and I consider that some points need to be reviewed:
Ln 47-48: Is correct the percentage of 3 – 15% for infection rate in revision TKA? The most recent study you cited (Yaghmour, K.M.; Chisari, E.; Khan, W.S. Single-Stage Revision Surgery in Infected Total Knee Arthroplasty: A PRISMA Systematic Review. J Clin Med 2019, 8, doi:10.3390/jcm8020174.) reported in the introduction that “The reinfection rate (RR) of single-stage revision surgery (5–25%) is comparable to two-stage revision (9–20%)”, while in the results they proved that “the reinfection rate ranged from 0–38%, with a mean of 15.42% (Median 15.00%, SD ± 10.42).” Also other more recent studies (e.g.: Reinfection and re-revision rates of 113 two-stage revisions in infected TKA. Joris Bongers, Anouk M.E. Jacobs, Katrijn Smulders, Gijs G. van Hellemondt, and Jon H.M. Goosen. doi: 10.7150/jbji.43705) reported higher infection rates.
Please, explain more clearly and specifically the aim of your study in the introduction.
Ln 82-84: You should be more specific for the method of exclusion of periprosthetic joint infection.
In the analysis of systemic blood sampling, you should report the negativity of Serum markers like C-Reactive Protein, D-dimer and Erythrocyte Sedimentation Rate, while in the analysis of joint aspiration you should mention the synovial biomarkers like alpha-defensin, leukocyte esterase, interleukin-6 and C-Reactive Protein.
Ln 90: Please describe better the indications for the TKA revision. Furthermore, did you perform any radiological examination?
Ln 102: Why did you not give any postoperative antibiotics? Usually, the standard postoperative prophylactic antibiotic after TKA revision includes postoperative intravenous first-generation cephalosporin every 8 hours for three times (e.g.: Extended Postoperative Prophylactic Antibiotics with First-Generation Cephalosporin Do Not Reduce the Risk of Periprosthetic Joint Infection following Aseptic Revision Total Knee Arthroplasty. Feng-Chih Kuo, Po-Chun Lin, Kerri L. Bell, Jih-Yang Ko, Ching-Jen Wang, Jun-Wen Wang. doi: 10.1055/s-0039-1683889).
Did you use tranexamic acid during the surgical procedure?
Ln 155: You reported that there were 57 positive samples, but the sum is different (2+13+16+13+4+8). Please correct the data.
Your study showed a microbial finding of irrigation fluid in 77,8% after 90 minutes, but no periprosthetic joint infection was observed within the follow-up period. Therefore the bacterial colonization of irrigation fluid does not represent a predictor for higher risk of articular infection, thus this evidence present low clinical relevance.
You should justify the accomplishment of this work in relation to your previous study (Contamination of Irrigation Fluid During Primary Total Knee Arthroplasty. Michael Fuchs, Philipp von Roth, Tilman Pfitzner, Sebastian Kopf, Frauke Andrea Sass, Hagen Hommel. doi: 10.5435/JAAOSGlobal-D-17-00027). The two studies differ for the surgical procedures (primary total knee arthroplasty versus aseptic revision knee arthroplasty), but they share the same objective, similar results and comparable conclusions. Thus, please analyze more precisely the differences between the obtained results regarding the time points of positive samples, and the differentiation of bacterial strains, because the percentage of detected pathogens seems to be very different between the two studies.
Round 2
Reviewer 2 Report
My concerns have been addressed.
i think it’s fine for publication.
Reviewer 3 Report
My concerns have been addressed.